# Country Distance and Entry Mode Choice of MNEs in Vietnam's Agricultural Sector in Context of Free Trade

**Nguyet Nguyen \*, Ha Thi Hoang Tran and Tuan Duong Vu** 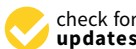

Business Administration Faculty, Thuongmai University, Hanoi 100000, Vietnam;
hoanghaqtcb@tmu.edu.vn (H.T.H.T.); vutuanduong@tmu.edu.vn (T.D.V.)
\* Correspondence: mynguyet@tmu.edu.vn; Tel.: +84-977-262-323

**Abstract:** In this article, we focus on tackling a relative research gap: how country distance (institutional, cultural, economic, and geographical distance) determines the entry mode choice between wholly-owned enterprises (WOEs) and joint venture enterprises (JVEs) in the context of "going global". Based on a sample of 439 multinational enterprises (MNEs) from 22 different nations that directly invested in the agricultural sector of Vietnam in the period 1996–2019, an empirical investigation has been conducted by employing logistic regression. The results show that as cultural and geographical distances increase, MNEs prefer JVE forms. However, WOE becomes more popular in cases of large economic and institutional distance. Furthermore, entry mode choices of MNEs are also noticeably impacted by freedom of trade.

**Keywords:** country distance; entry mode; Vietnam's agriculture sector; free trade; FTAs

## 1. Introduction

The trend of globalisation has pushed investment activities abroad and the formation of multinational enterprises (MNEs). In the framework of internationalisation decisions, the selection of entry mode is an important matter because it determines the degree of control over the activities of enterprises in foreign markets [1]. A suitable market entry mode will help MNEs create competitive advantages and even determine investment efficiency and development in the host country [2]. Therefore, it is necessary to learn about the factors that affect the investment form of businesses. Several studies on international business have highlighted how international expansion involves overcoming barriers [3,4]. The investment decision also depends on the distance between the home country and the host country [5,6]. However, the results are ambiguous, resulting in the necessity to perform further studies on this topic. Moreover, previous studies have focused mainly on the choice of the form of investment by MNEs in general, and there have not been many forms of research associated with specific industries or fields of investment attraction, especially in the context of trade liberalisation and the introduction of Free Trade Agreements (FTAs). Several studies have also found that FTAs are associated with higher foreign direct investment (FDI) flows; extant work tends to ignore variation within FTAs and among types of FDI [7]. Models of the effects of political risk on firms' preferences for high-control modes (i.e., green-field investments or mergers and acquisitions) versus low-control modes (i.e., joint ventures or licensing agreements) suggest that FTAs could have a substantial impact on entry-mode decisions [8]. From the above arguments, it is necessary for more studies on the influence of country distances on the choice of FDI entry mode of MNEs in economic integration. This study will analyse MNEs in Vietnam's agricultural sector for the following reasons.

Firstly, although Vietnam has been progressively promoting the development of the industrial and services sectors, agriculture still plays a critical role in the economy. Agriculture contributes approximately 20% of GDP and creates jobs for 60% of the population

living in rural areas. Furthermore, agriculture is also the pedestal or foundation of Vietnam's economy [9]. However, although FDI inflows into Vietnam's other industries have increased sharply, FDI inflow into Vietnam's agriculture sector has been minimal, representing only 1.7% of the total number of FDI projects and 1% of the total FDI capital into Vietnam. This goes against the trend of FDI into other sectors of Vietnam and goes against the world's FDI inflows into agriculture [9]. As a result, the attraction of FDI into this industry should be given due attention and appropriate policies should be developed to attract investment. It is, therefore, necessary to reduce the country' distance to encourage different entry modes in the agricultural sector in Vietnam.

Secondly, Vietnam has experienced a significant economic transition process, but there are still weaknesses in the country's formal and informal institutions, which remain significant obstacles for conducting business [10]. Despite the positive changes to investment that the Enterprise Law introduced in 2020, the levels of competitiveness and international integration of local counterparts in Vietnam still differ vastly from those of developed economies. Hence, the type of relationships established by foreign firms with local partners in these economies also differed widely [11]. Economic reforms had been implemented in the transition process, but institutions and cultural reforms were rejected [12]. Such a situation bears additional uncertainties and complexities that can affect FDI flow in agriculture in Vietnam, where the risks are higher than for other investment sectors.

Thirdly, in terms of investment in agriculture, there are currently two primary forms: wholly-owned enterprises (WOEs) and joint venture enterprises (JVEs). WOE is a form in which MNEs acquire the right to use agricultural land to operate production and business and have complete control over farming activities. The landowner does not have the power to make decisions regarding the management and operation of the investment company. This is a globally popular form of investment in countries with developed agriculture. It provides means to improve production capacity and management of agricultural production with high qualification via the lengthened arm of market transactions. This helps produce safe, high-quality agricultural products with lower production and marketing costs. In developing countries, WOE can help overcome the imperfectness of input and output markets and handle capital, seed, machinery, equipment, and market [13]. The JVE model in the agricultural sector is a form of investor participation whereby two or more parties jointly run a business. Each party contributes to business activities in cash (capital) or in kind (rights to land, resources, technology, and know-how) and is responsible for the business results of the enterprise. Joint ventures are formed at different degrees, in which it is common to establish a consolidated or jointly owned company and an agreement on the shares of the parties involved, which allows the joint venture to be subjected to limited liability and property ownership. In essence, agriculture is a potentially risky industry and a joint venture is considered an attractive form because it is an association between small businesses that are mobilised to contribute to joint assets (technology transfer or land lease), share costs (damage, land rent, plant seeds, baby animals, and production materials), share profits, and to make business management decisions together. The benefits of JVEs are clear commitments, reduced political risks, and help in building and developing brands [14]. However, over 70% of investment projects in the agricultural sector are in the form of WOEs and the number has tended to increase over time.

Finally, Vietnam has a high level of international economic integration. This is a result of the active participation and negotiations of FTAs with other countries, especially new-generation FTAs which have been negotiated and signed in recent times and contain enormous scope and a high level of liberalisation. As of June 2020, Vietnam officially joined 13 FTAs (seven of which were signed as a member of ASEAN and six of which were signed as an independent party). In addition, Vietnam is currently negotiating three FTAs. Vietnam has FTA partnerships with 56 economies (fifty-five countries and one territory). Vietnam's signing of bilateral and multilateral FTAs has enabled Vietnamese enterprises to expand their markets and gain access to regional and global markets [15]. FTAs have made positive improvements in attracting foreign investment in Vietnam and have helped to

enhance administrative institutions and business environments. This paper aims to explain the relationship between country distance and entry-mode choice in Vietnam's agriculture sector.

## 2. Literature Overview and Research Hypothesis Development

### 2.1. Entry Mode

The internationalisation process of enterprises is divided into different stages, including the choice of investment location, mode of entry, degree of ownership, and level of capital control in the foreign market [16]. Among these stages, the selection of entry mode is an important decision because it is the method used by enterprises to penetrate foreign markets [17]. More specifically, it is an agreement on how to transfer a business's skills, technology, know-how, and resources in a foreign market [18]. When deciding to serve new markets, MNEs determine the degree of ownership of their operations in overseas markets [19]. In the international business literature, entry modes are classified as non-equity-based (indirect investment such as export and contractual agreements) and equity-based (direct investment) [20]. In the equity-based entry mode, MNEs may select wholly-owned subsidiaries (WOSs) or share ownership with partners by forming joint ventures with majority, equal, or minority control [19]. Vietnamese law of investment categorised two forms of FDI: 100% foreign-invested, wholly-owned enterprises (WOE), and joint venture enterprises partially owned through capital contribution or share acquisition (JVE) [21]. Each entry mode is defined by different control, ownership, risk levels, and commitments of resource-and-information control [22]. However, this form requires more time to set up the business and more complex information requirements [23]. In terms of control, commitment, and risk, WOE is selected when an MNE desires to achieve the highest level of control based on making the highest level of commitment and adopting the highest level of risk. On the other hand, MNEs prefer WOE since it provides MNEs with an opportunity to expand quickly independently of a partner; thus, it is unnecessary to cooperate with a partner whose decisions and behaviour rules are not well known or understood by the foreign firm. JVE allows the transfer of strategic assets from the headquarters to the MNEs to have a higher level of control over their foreign subsidiaries, thereby gaining full benefits from these subsidiaries and it obtains a lower level of control but the level of risk is reduced [24]. JVEs reduce the investment and commitment of resources such as equity capital and the opportunity costs of managerial talent sent to run the foreign business. Several studies on the determinants of foreign investment choice, including the transaction cost theory [5,17], have emphasised that the selection of entry mode was considered based on the predicted costs that firms must pay to enter the foreign market. Therefore, it results from a decision-making process that compares the costs and cost amendments associated with alternatives. This means entry-mode choice, which helps minimise transaction costs. However, transaction costs are related to many factors, such as differences in culture, economic development, or institutions. Hence, choosing international market participation involves overcoming barriers to such participation. Institutional theory [25] emphasises that the institutional environment is a determinant of a firm's behaviour and structure, as it must adapt to the local regulations of host countries. Institutional theory requires MNEs to understand the nature of the institutional environment and overcome institutional differences to adapt to the local market. The eclectic theory [26] proposes that an enterprise's selection of investment forms is based on factors such as ownership advantage, location advantage, internalisation advantage, institutional conditions, and government intervention. MNEs face multiple institutional environments and, in order to maintain legitimacy, they have to comply with the different frameworks of each host country [27]. Therefore, this research combines two theories and empirical studies to better explain the factors referred to in our study.

### 2.2. Institutional Distance

Institution and institutional distance are topics commonly covered in studies on entry mode. Institutional distance reflects institutional dissimilarity between the host and home countries [27]. MNEs must adapt and conform themselves with the differences of the institutional environment to establish business legitimacy and to ensure subsequent business success [22]. Institutional distance is used to test if institutions affect entry mode [1]. Institutional distances are measured by factors such as country risk, legal distance (regulatory dimension), attitudes regarding requesting government benefits, and corruption. A governance indicator index contains six dimensions to measure institutional distance: Voice and Accountability, Political Stability and Absence of Violence, Government Effectiveness, Regulatory Quality, Rule of Law, and Control of Corruption [28]. The criteria include legal distance, attitudes and government interest requirements towards international business, and corruption. From an economic perspective, institutional distance amplifies the difficulty and cost consumption of the practical transfer of home-based internal resources, procedures, and management practices to the host country [29]. The MNEs know better in terms of how to cope with corrupt government officials, political issues, and institutional voids and, thus, choose JVE. In contrast, MNEs in developed countries invest in the transition economy using WOE if institutional distances are high [12]. However, studies with samples of MNEs from developed countries that internationalised to emerging countries [30] and MNEs from emerging countries that moved indicate that those firms preferred WOE when the cultural distance between the home and host country was high. Based on these arguments, the study formulates the following hypothesis.

**Hypothesis 1:** *The greater the institutional distance between home and host countries, the greater the probability that MNEs in the agricultural sector will choose WOE over JVE.*

### 2.3. Cultural Distance

Cultures are shared values and beliefs that differentiate groups of people from each other. They are values shared in common that shape a group of people [30]. Cultural distance is the differences in values and beliefs between home and host country. There is no consensus among scholars regarding the influence of cultural distance on the choice of entry mode by MNEs. Some studies suggested that the more considerable the cultural difference is, the more significant the organisational and management differences will be between enterprises and the higher the transaction costs will be for MNEs [31,32]. Increasing cultural distance leads to external uncertainty related to complications and, thus, results in high adoption costs associated with WOE. The uncertainty from the external environment also increases; in this manner, the WOE model is more severe than others. These results indicated that companies should choose JVE [33] because JVE helps MNEs share benefits and risks with local businesses while reducing the uncertainty from a culturally different country.

Additionally, JVE demands fewer resources; thus, it has lower exit costs. However, some opposing views believe that, in the context of rising cultural distance, MNEs favour WOE [34,35]. In this case, it is difficult for MNEs to integrate into joint venture networks in countries with cultural differences. Moreover, the fact that MNEs come from different cultures can raise some issues in integration among employees in a JVE business. On the other hand, MNEs will find it easier to integrate into the mode of WOE as enterprises have the opportunity to set up an organisational structure model, transfer technology, and apply management methods from scratch without having to accept existing methods or to find how to select models and staff that are appropriate to their national culture [30]. From the above explanations, we propose the following research hypothesis.

**Hypothesis 2:** *The greater the cultural distance between home and host country, the greater the probability that MNEs in the agricultural sector will choose JVE over WOE.*

### 2.4. Economic Distance

Economic distance is defined as the difference in the levels of economic development between countries [36]. Differences in economic levels can be reflected by energy purchases, labour costs, macroeconomic stability, or the degree of openness of economies [37]. The economic range offers opportunities for MNEs to explore and exploit their respective resources, and this factor can determine the choice of entry mode [38]. When MNEs in countries with more developed economies wanted to transfer advantages, they tended to choose WOE, while MNEs with weak competitive advantages often chose JVE in order to gain greater access to strategic assets [24]. Supporting this point of view, the authors of [39] found that investors choosing JVE in a field where the investor-owned technology superior to that of the host country will choose WOE on the contrary. In developing countries with low technological capacity, investment companies from developed economies will have the opportunity to effectively exploit technological advantages. Therefore, the research hypothesis formed here can be stated as follows.

**Hypothesis 3:** *The greater the economic distance between home and host country, the greater the probability that MNEs in the agricultural sector will choose WOE over JVE.*

### 2.5. Geographical Distance

Geographical distance refers to the distance in terms of geographical location, as well as to the transport and communication infrastructure disparity between the two countries [36]. The measurement of geographical distance between countries is usually performed by determining the distance between the two main cities in each country. According to international business theory, geographical distance negatively affects trade between countries [40]. In favour of this view, the transaction cost theory argues that large geographical distances raise transportation and communication costs and even increase the time of delays, which causes a competitive disadvantage [41]. Indeed, in terms of transport distance, neighbouring countries prefer WOE because it reduces transportation costs and coordinates the movement of goods or resources between countries [37].

On the contrary, the rise in transport distance entails a cost burden and MNEs should find partnerships to share this cost when accessing distant markets. From an information perspective, long geographical distance worsens information asymmetry and reduces the ability to receive accurate and updated information from the counterparty, thus enhancing the uncertainty. This explains why MNEs prefer JVE to WOE [42]. Therefore, the research hypothesis is built as follows.

**Hypothesis 4:** *The larger the geographical distance between the investing country and the host country, the greater the probability that MNEs in the agricultural sector will choose JVE over WOE.*

### 2.6. Investment Size

Investment is the next factor mentioned in the selection of entry mode of MNEs. It is defined as the assets or start-up capital that MNEs transfer into the host market [1]. According to transaction cost theory, the larger the size of the invested capital, the higher the costs of entering and exiting the market, as well as with respect to financial and operational risks [43]. Moreover, enterprises with little capital need to implement JVE because they do not want to share advanced knowledge and technology with business partners and, in this way, they can limit partners' opportunistic behaviour. Thus, the more significant the investment size, the better the MNE will be able to deal with opportunism and the higher the probability that costs will be minimised; as a consequence, the likelihood of complete control by WOE is larger [44].

On the other hand, MNEs with larger investment sizes are more likely to have diverse resources which can be applied effectively to new market entry, and larger firms have a

greater capacity to establish their own operational business in a foreign country. Therefore, project size is considered to positively influence the choice towards WOE [43]. From the above arguments, the following hypothesis is proposed.

**Hypothesis 5:** *The greater investment size, the greater the probability that MNEs in the agricultural sector will choose WOE over JVE.*

*2.7. Trade Freedom*

Free trade is referred in this study as comprising factors such as host country tariffs and compliance costs of importing and exporting, as well as control over the movement of capital and people across borders [45]. One of measures in expanding international trade liberalisation is signing free trade agreements (FTAs) between at least two countries to reduce trade barriers and promote trade in goods, services, and investment between those countries. It is considered a national agreement that seeks to prioritise trade or services between two or more countries [46]. Accordingly, countries will follow the roadmap of reducing and eliminating tariff and nontariff barriers to establish a free trade area. The FTA is a commitment with a high degree of credibility and it demonstrates the determination of governments towards market opening and free trade. This reduces investment risks and facilitates the reduction in costs and information asymmetries for multinationals when they enter the market [45]. In addition, a distinction of the FTA is that the commitments in the framework of the agreement affect and bring benefits to the participating countries and signal the presence of a more favourable investment environment to third parties. It encourages companies to move capital flows to these countries [47]. According to the theory of transaction cost, in countries with low levels of risk and policy uncertainty, MNEs prefer WOE to JVE [48]. There is a long-run equilibrium relationship between trade openness and FDI for the economy and its sectors, and government policies should focus not only on promoting the level of trade openness in the economy but also on the magnitude of the degree of openness in agricultural and industrial sectors [49]. From the above arguments, the following research hypothesis is proposed.

**Hypothesis 6:** *In the condition that the investment country and the host country have signed an FTA, the probability that MNEs in the agricultural sector will choose WOE over JVE is greater.*

## 3. Methods

*3.1. Research Sample*

The study uses a dataset of 704 MNEs invested in the agricultural sector of Vietnam in the period 1996–2019, provided by the Ministry of Agriculture and Rural Development (2020), with information on the mode of entry, representatives, country, line of business, and revenue. The identification of selected enterprises for analysis is based on the following: (i) enterprises investing in the industry from 1996 (After the Law on Foreign Investment took effect) to 2019 (before the outbreak of the COVID-19 epidemic); (ii) enterprises for which its ownership structure changed by no more than 10% during the studied period; (iii) companies investing in the agricultural sector through WOE and JVE; and (iv) investing countries with data on the Hofstede and WGI cultural index. Finally, the sample consists of 439 enterprises from 22 different countries included in the analysis (Table 1), in which the sample structure includes FDI enterprises in the agricultural sector of Taiwan (21.18%), Korea (10.8%), Japan (10.8%), Thailand (7.52%), China (6,8%), France (6.38%), Singapore (6.15%), and other countries.

**Table 1.** FDI in Vietnam agriculture by investing country.

|  | Country | Samples | Percentage |  | Country | Samples | Percentage |
|---|---|---|---|---|---|---|---|
| 1 | India | 6 | 1.37% | 12 | Japan | 46 | 10.48% |
| 2 | Australia | 23 | 5.24% | 13 | France | 28 | 6.38% |
| 3 | Belgium | 1 | 0.23% | 14 | Philippines | 2 | 0.46% |
| 4 | Canada | 4 | 0.91% | 15 | Singapore | 27 | 6.15% |
| 5 | Taiwan | 93 | 21.18% | 16 | Thailand | 33 | 7.52% |
| 6 | Netherlands | 14 | 3.19% | 17 | Switzerland | 2 | 0.46% |
| 7 | South Korea | 46 | 10.48% | 18 | China | 28 | 6.38% |
| 8 | Indonesia | 1 | 0.23% | 19 | United Kingdom | 3 | 0.68% |
| 9 | Malaysia | 18 | 4.10% | 20 | British Virgin Islands | 20 | 4.56% |
| 10 | United State | 15 | 3.42% | 21 | Hong Kong | 21 | 4.78% |
| 11 | Russia | 7 | 1.59% | 22 | New Zealand | 1 | 0.23% |

*3.2. Research Scale*

- Dependent Variable

Entry mode (EoM): The research selects the dependent variable to be a form of FDI in the agricultural sector with two types: WOE (full ownership) and JVE (partial ownership). Information about the form of investment is displayed on the investment registration license for the first time in Vietnam and is conventionally valued at 1–WOE and 0–JVE [5].

- Independent Variables

Institutional distance (InDis): Institutional distances are measured on six dimensions, including voice and accountability, political stability, and lack of accountability; political stability and absence of violence; government effectiveness; regulatory quality; the rule of law; and control of corruption, collected from the WGI index [28].

$$ID_{aj} = \sum_{i=1}^{n} \left\{ \frac{(I_{ia} - I_{ij})}{V_t} \right\} / 6$$

In this formula, $ID_{aj}$ reflects the institutional distance between two studied countries, host (*a*) and home (*j*); $I_{ia}$ is the value for the home country *j*; and $I_{ij}$ is the value for the host country. Variance in the equation is $V_t$. The total is divided by 6. This study uses the Worldwide Governance Indicators (WGIs) provided by the World Bank during 2016–2019 in [50].

Cultural distance: The cultural distance is reflected in six aspects of Hofstede's culture [30], including the following: Power, Individualism, Masculinity, Uncertainty Avoidance, Long-term Pragmatism, and Indulgence. Scores of aspects ranging from 0 to 100 were collected in [51] and calculated by the formula of Kogut and Singh [5] as follows.

$$CD_{aj} = \sum_{i=1}^{n} \left\{ \frac{(I_{ia} - I_{ij})}{V_t} \right\} / n$$

In this formula, $CD_{aj}$ reflects the culture distance between countries host (*a*) and home (*j*); $I_{ia}$ is the value for the home country *j*; and $I_{ij}$ is the value for the host country. Variance in the equation is $V_t$. The total is divided by $n$, a value of 4 is used if the host countries invest in Vietnam before 2010, and a value of 6 is used if the investment is established after 2010. The Hofstede cultural index is collected in [51].

Economic distance (EcoDis): The economic difference is calculated as the difference of GDP per capita (unit as USD 1000) between the investing countries and Vietnam at the time of investment. The index of GDP per capita is accessed in [52]. This study uses the natural logarithm of absolute difference in GDP per capita index between the home country and Vietnam.

Geographical distance (GeoDis): It is calculated as the distance (calculated unit as 1000 km) between two countries' capitals. The index of geographical distance between Vietnam and investing countries was collected in [53]. The natural logarithm of the great circle distance between Vietnam and the capital city of MNE's home country in kilometers is used in this study.

- Control Variable

Investment size (Ven): the determination of this index is based on the total registered investment capital of the project at the time of the first investment license. The size of the investment capital is measured in millions of USD, and we use its logarithm form. This data were collected from the database in [54].

Free Trade Agreements (FTAs): The participation in FTAs by investing countries and Vietnam is determined based on the effective date of 13 FTAs that our country has signed. When investing in Vietnam's agricultural sector, the investing countries that have entered into an FTA with Vietnam have a value of 1, and those that have not joined an FTA with the host country have a value of 0. Data on FTAs of Vietnam and member countries are provided in [15].

*3.3. Analytical Methods*

The logistic regression method was performed by IBM SPSS 22 software to evaluate the influence of country distance on the selection of entry mode in agriculture as the dependent variable has two values: 1 corresponding with WOE and 0 corresponding with JVE [18]. This technique is suitable for considering the relationship between independent and dependent variables by evaluating the probability of choosing WOE or JVE by the enterprise instead of the independent assessment. The regression coefficient estimates the impact of institutional, cultural, geographical, and economic variables on the probability of choosing an investment choice. The positive correlation coefficient shows that the independent variables tend to increase the probability of choosing WOE over JVE and vice versa. The purpose is to evaluate the appropriateness of the research model by checking the model's errors, such as multicollinearity (VIF coefficient), autocorrelation, and variable variance.

## 4. Results

*4.1. Descriptive and Correlation Analysis*

The results of descriptive statistical analysis (Table 2) indicated that foreign direct investment in Vietnam's agricultural sector over the past three decades has mainly come from 22 countries around the world divided into five central regions: East Asia (53.3%), Europe (17.08%), North America (4.33%), Oceania (5.69%), and Southeast Asia (22.1%). In these statistics, the most popular home countries of MNEs in the agricultural sector were Taiwan (21.18%), Korea (10.8%), Japan (10.8%), and China (6.38%). This makes sense because Korea and Japan have been Vietnam's partners with the most significant foreign direct investment in the last over 30 years of Doi Moi (reform). Second to Korea and Japan are some regional countries such as Thailand (7.52%), France (6.38%), and Singapore (6.15%). ANOVA analysis on the difference of investing regions shows that East Asia has the lowest cultural distance with Vietnam (1.38), while Oceania has the highest value (2.96). The enormous institutional difference belongs to Oceania countries (7.64) and the lowest in Southeast Asia (3.68). The economic distance between Vietnam and the countries in North America is the largest (10.0), while the distance with Southeast Asian countries is the lowest (9.05). Finally, the furthest geographical distance to Vietnam is in Europe and Southeast Asia (9.00) and the lowest is in East Asia (7.39). In addition, at the time of investment, 11 countries from East Asia, Oceania, and Southeast Asia have signed bilateral or multilateral FTAs with Vietnam.

**Table 2.** FDI in Vietnam agriculture by investing regions.

| Regions | CulDis | InDis | EcoDis | GeoDis |
|---|---|---|---|---|
| East Asia (*n* = 234) | 1.38 *** | 4.09 *** | 9.32 *** | 7.39 *** |
| Europe (*n* = 75) | 2.53 *** | 6.73 *** | 9.71 *** | 9.00 *** |
| North America (*n* = 19) | 2.79 *** | 6.89 *** | 10.00 *** | 9.00 *** |
| Oceania (*n* = 25) | 2.96 *** | 7.64 *** | 9.08 *** | 8.96 *** |
| Southeast Asia (*n* = 86) | 0.63 *** | 3.68 *** | 9.05 *** | 7.60 *** |

N = 439, *** $p < 0.001$.

NEs investing in agriculture chose the WOE investment form (77.9%), which outnumbered those that chose JVE (22.1%). MNEs in Vietnam agriculture sector are scattered in seven agricultural economic regions of the nation, of which most of the focus is on the Southeast region (26.42%), the South Central Coast (18.00%), the Central Highlands (16.17%), and the Mekong River Delta (14.81%) because these are areas have a tradition of agricultural development, favorable traffic conditions, high level of intensive farming and machine application, commodity-oriented agricultural production, and heavy use of agricultural machinery and materials. After the IX National Assembly, the 10th session, approved the Law on Foreign Investment in Vietnam on 12 November 1996, foreign direct investment in Vietnam's agricultural sector began to develop. Since then, the history of the development of FDI in the agricultural sector has been divided into three periods: (i) 1996–2000; (ii) 2001–2010; and (iii) 2011–2019. The results of the ANOVA analysis by investment stage (Table 3) revealed that, in the first stage, there was a market entry by 70 MNEs (15.95%); in the next period, 2001–2010, a substantial increase in FDI projects was recorded in the agricultural sector with 180 projects (accounting for 41.0%) and a slight increase in the number of projects in the period 2011–2019 remained stable with 189 projects (accounting for 43.05%). The Sig results of the F-test (0.001 and 0.026, respectively, are less than 0.05) indicate a difference in economic distance and geographical distance. Specifically, at the initial stage of attracting foreign investment, investment flows into the agricultural sector mainly came from countries with short geographical distances and small economic distances compared to Vietnam. In the next phase, this sector began to attract capital flows from countries outside the region and developed countries globally.

**Table 3.** FDI in Vietnam's agricultural sector by investment stage.

| | CulDis | InDis | EcoDis | GeoDis |
|---|---|---|---|---|
| 1996–2000 (*n* = 70) | 1.46 | 4.74 | 9.01 ** | 7.67 * |
| 2001–2010 (*n* = 180) | 1.59 | 4.91 | 9.32 ** | 7.84 * |
| 2011–2019 (*n* = 189) | 1.62 | 4.64 | 9.51 ** | 7.97 * |

N = 439, * $p < 0.05$, ** $p < 0.01$.

Table 4 provides descriptive statistics and correlation analysis, showing that the correlation coefficients of all variables are less than 0.7. Furthermore, the VIF coefficients' values are all less than 2.0, and the VIF coefficient of the institutional distance variable reaches 2.03> 2.0. However, the value of VIF value is considered a problem in regression analysis only when it is more than 10, normally a VIF of 5 or above, indicating that multicollinearity problems can appear [55]. Therefore, it can be concluded that there is no multicollinearity in this study.

**Table 4.** Correlation analysis results.

| | Mean | Std. E | VIF | EoM | InDis | CulDis | GeoDis | EcoDis | Ven | FTA |
|---|---|---|---|---|---|---|---|---|---|---|
| EoM | 0.78 | 0.415 | | 1 | | | | | | |
| InDis | 4.77 | 2.684 | 2.030 | 0.041 | 1 | | | | | |
| CulDis | 1.58 | 1.113 | 1.778 | −0.190 ** | 0.356 ** | 1 | | | | |
| GeoDis | 7.87 | 0.809 | 2.235 | −0.155 ** | 0.551 ** | 0.644 ** | 1 | | | |
| EcoDis | 9.35 | 0.993 | 1.569 | 0.205 ** | 0.578 ** | 0.141 ** | 0.376 ** | 1 | | |
| Ven | 7.77 | 1.478 | 1.020 | 0.093 | 0.094 * | −0.023 | −0.018 | 0.006 | 1 | |
| FTA | 0.41 | 0.492 | 1.085 | 0.082 | −0.234 ** | −0.137 ** | −0.072 | −0.058 | −0.014 | 1 |

N = 439, * $p < 0.05$, ** $p < 0.01$.

### 4.2. Logistic Regression Analysis Result

The mode of entry is a dependent variable with two values attached; thus, the study uses binary logistic regression to test hypotheses according to two models: (1) the model of independent variables and (2) the overall model with both independent and control variables. The results of assessing the relevance of the research model (Table 5) by using the value of −2Log-likelihood in the studied models tend to decrease significantly, indicating that the regression results are promising [56]. In addition, the value of Pseudo R Square, built based on R-Square in the model of binary logistic regression [56] including Cox and Snell [57] and Nagelkerke [58], has attained a significant improvement from model 1 to model 2, and the prediction rate increased from 78.4% to 79.5%, which shows the suitability of the logistic regression model. Therefore, it can be confirmed that model 2 fits the data well and was selected for analysis.

**Table 5.** Results of testing binary logistic regression.

| Entry Mode (1-WOE)/(0-JVE) | Mode 1 | | Mode 2 | |
|---|---|---|---|---|
| | β | Exp(B) | β | Exp(B) |
| Institutional distance | 0.105 (0.118) | 1.058 | 0.135 (0.061) | 1.145 |
| Cultural distance | −0.403 ** (0.014) | 0.697 | −0.450 * (0.010) | 0.638 |
| Geographic distance | −0.651 ** (0.003) | 0.549 | −0.623 ** (0.005) | 0.536 |
| Economic distance | 0.700 *** (0.000) | 2.088 | 0.745 *** (0.000) | 2.106 |
| Investment size | | | 0.173 * (0.037) | 1.189 |
| FTA | | | 0.599 * (0.032) | 1.821 |
| Constant | 0.208 | | −2.549 | |
| *−2 Log Likelihood* | 410.494 | | 401.797 | |
| *Chi-Square* | 33.727 | | 10.075 | |
| *% Correct* | 78.4 | | 79.5 | |
| *Pseudo R Square* | 0.114 | | 0.131 | |
| *Nagelkerke R2* | 0.175 | | 0.202 | |

N = 439, * $p < 0.05$, ** $p < 0.01$, *** $p < 0.001$. Please find more information in the Supplementary Materials.

Hypothesis 1 suggests that the larger the institutional distance between the home and host countries, the more commonly MNEs investing in the agricultural sector will choose WOE rather than JVE. The estimated results reveal β = 0.135 and sig = 0.061 < 0.1, which means that the large institutional difference has a positive influence on the selection of WOE, and this hypothesis is supported. Hypothesis 2 states that cultural distance negatively impacts the choice of entry mode for WOE rather than JVE, which means that when the cultural distance between Vietnam and the home country increases, the tendency to choose WOE becomes less popular. The cultural distance model has β = −0.450 and

sig = 0.010 < 0.05, meaning that this hypothesis is accepted. Hypothesis 3 assumes that geographical distance negatively influences the choice of the investment of WOE than JVE. The analysis results indicate that geographical distance has the coefficient b = −0.623 and sig = 0.005 < 0.05, which proves that the larger the distance between Vietnam and the home countries, the tendency to choose WOE will decrease and this hypothesis is supported. Hypothesis 4 suggests a positive relationship between economic distance and entry mode choice. As the economic distance between the home countries and Vietnam increases, MNEs tend to choose WOE rather than JVE. The analysis results reveal that coefficient β = 0.754 and sig = 0.000 < 0.05 support this hypothesis. Hypothesis 5 of the influence of the investment size on the choice of WOE compared with JVE results in β = 0.173 and coefficient Sig = 0.037 < 0.05. This means the more significant the investment size, the more likely the enterprise chooses WOE over JVE, and this hypothesis is accepted. Finally, hypothesis 6 of FTAS between Vietnam and the home country having a positive influence on the choice of investment form observes the results of β = 0.599 and Sig = 0.032 < 0.05; for this reason, this hypothesis is supported.

## 5. Discussion

The study examines the impact of country distance on the choice of entry mode by MNEs in the agricultural sector in Vietnam. The specific results of the study are as follows:

First, a more considerable institutional distance has a more positive impact on the choice of WOE than JVE by MNEs in the agricultural sector in Vietnam at the significance level of 10%. This result is similar to the conclusions in the studies of [12,59]. Investment in the agricultural sector is influenced by many factors, such as competitive markets, formal and informal institutions, and macro policies. The government and local agencies play an essential role in developing and perfecting the system of laws and regulations and implementing operational supervision to protect the interests of relevant parties. The enhancement and development of the investment environment and strengthening position as a major attraction for foreign investments at the regional and global levels are performed by the following: expanding and diversifying the production base, facilitating the registration and licensing of investment projects, activating the trade movement and opening new export markets for local industries, and providing some incentives for the foreign investments, such as freeing some taxes and agricultural land pricing to encourage investment [9].

Furthermore, investment procedures must be oriented towards convenience, supporting mechanisms for the involved parties as well as aiming to reduce transaction costs. The JVE form also requires strong support from the government's policy frameworks such as consulting services, brokerage, project construction support, and risk guarantee. However, there are still too many overly strict regulations regarding land, significantly affecting the process of land accumulation and even making conditions conducive to corruption. The land policy is still inadequate, resulting in difficulties for foreign investors to access land for agricultural production. For example, there are many restrictive regulations on land use rights and transfer of use rights, land-area entitlement to each household, and change of land use purpose. These regulations are meant to ensure equal access to land. However, they limit the ability to accumulate land and hinder long-term investment and large-scale production.

Second, cultural distance negatively influences the choice of WOE versus JVE by MNEs in Vietnam's agricultural sector. The findings indicated that when the cultural differences between Vietnam and the home countries became more significant, MNEs tended to choose JVE. This result was consistent with those of the studies by [32,60], who argued that a considerable cultural distance would result in an increase in prices and costs associated with contracting, negotiation, and monitoring and would create more risks, which in turn would encourage MNEs to choose JVE. However, this result contradicts the research conclusion of [61], who found a positive impact of cultural distance on WOE choice in Vietnam. Agriculture is a traditional field in Vietnam with long-standing customs and

farming practices and where attention is often paid to experience and habits. Agricultural practices have existed for a long time and are not easily changed in a short period. The practice of small, fragmented production, lack of links, and mutual assistance will be a significant obstacle hindering agricultural production from developing into large commodity production. Changing agricultural production habits towards a commodity-based mechanism with technology application requires a great deal of time and training costs. Therefore, JVE allows MNEs to penetrate the culture, make changes, and transfer production methods to the workforce in a manner that suits the culture. In addition, agriculture is an industry that produces raw materials and is closely related to the environment. JVE is a form of investment that enables MNEs to transfer technology as well as find raw materials at a low cost.

Third, similarly to cultural difference, geographical distance negatively influences the choice of WOE in the agricultural sector in Vietnam. This result is consistent with the studies of [36,37], who argued that the farther apart the countries are, the higher the logistics costs and the more challenging the difficulties in communication, making JVE a suitable form of investment in this case. It is significant in the agricultural industry when the output products of the industry require a strict preprocessing and preservation regime to ensure quality. At the same time, the transportation of agricultural products also requires the satisfaction of specific technical standards, and to this end, the longer the geographical distance, the greater the costs and risks borne by MNEs. Indeed, most MNEs in the agricultural sector aim to minimise risks in sourcing raw materials against weather, climate change, and markets. Meanwhile, the current infrastructure conditions of Vietnam's agricultural sector have not yet met the requirements of commodity-oriented agricultural production, especially the logistics system; therefore, consolidating the value chain by using JVE is a suitable option for investors.

Fourth, economic distance has a more positive impact on the choice of WOE than JVE in the agricultural sector of Vietnam. The finding is in line with prior research [24]. This change shows that the influence of Vietnamese economy development on the selection of projects is limited. With full authority to decide on the production and business activities of the enterprise, foreign investors will adjust investment activities in a direction that is the most beneficial for the enterprise. Furthermore, most MNEs in this sector have competitive advantage in technology, management, and operational experience; thus, the possibility of transferring to Vietnam business partners via JVE is even lower.

Fifth, the investment size in Vietnam's agricultural sector positively influences the choice of WOE form. The result for investment size is in accordance with [62]. This is attributed to the fact that agriculture investment requires a reasonable amount of investment capital in infrastructure, science and technology, research, and development of new products. On the other hand, investment in agriculture is a form of investment that has many risks resulting from being directly affected by weather, natural disasters, and epidemics. Therefore, many MNEs possessing technology strengths and significant capital tend to choose WOE to have the highest control over their operations and capital in the host country. This is beneficial for them to enter market via WOS since they can adjust and adapt quickly to the local environment. In addition, most Vietnamese agricultural enterprises have are small scale and limited competitiveness. Therefore, it is difficult for companies with large capital expenditures to find joint venture partners who can meet the requirements and purposes of the project.

Finally, the participation in bilateral and multilateral FTAs by Vietnam and the investing countries has a more positive influence on the choice of WOE as the investment form than JVE. Studies support this result. However, it contrasts with [62], who argued that joining an FTA could encourage more member countries to select JVE form because commitments under an FTA would reduce risks and improve spill-over effects to FDI in the form of JVE compared to WOE. Vietnam is a member of 13 FTAs, making Vietnam one of the economies in the region and globally with the most FTAs. Within the framework of these FTAs, there are commitments to the fair treatment of domestic investors and foreign

investors in establishment, acquisition, expansion, administration, deployment, operation, and business. However, most investment projects in the agricultural sector are hi-tech oriented. Therefore, technology transfer in the form of JVE is not the optimal solution because Vietnam's business environment still has some limitations. Furthermore, FTAs also play a role in moderating the relationship between the factors of institutional distance and the choice of WOE. Given trade liberalisation, countries participating in FTAs have improved institutional conditions, and investors perceive policy risks to be lower. This motivates MNEs to choose the higher control method, which is WOE [63]. Current FTAs provide general mechanisms for cooperation and investment attraction and specific mechanisms in investment activities in the agricultural sector. In addition, in joining FTAs, economic distance encourages MNEs investing in Vietnam's agricultural sector to choose JVE.

## 6. Conclusions

This study draws on the theories of transaction costs, institutional theory, and eclecticism theory to predict the trend of choosing investment forms of WOE and JVE in the agricultural sector of Vietnam in the context of economic integration and trade liberalisation. The findings have evidence of the influence of six studied variables included in the model. As the cultural distance and geographical distance between the host country and the home country are reduced, it enhances the tendency for JVE to be chosen, while institutional distance and investment size are believed to promote the popularity of WOE. Contrary to some studies on investment in general, economic distance positively affects WOE selection. The surprising result found in this study is the influence of FTAs on WOE selection. Although FTAs help reduce risks and trade barriers between countries while expanding the market, this factor still has a driving impact on the selection of WOE. However, study results reveal that, in the condition of significant cultural and institutional distance, there is an increased tendency to choose WOE. From these research results, some implications for the Vietnamese government in strengthening the attraction of foreign direct investment in the agricultural sector are given as follows.

First, it is necessary to combine large-scale projects that have an essential impact on the economy and agriculture of the region with small and medium-sized projects in difficult socioeconomic conditions to ensure the economic structure of regions and sectors.

Second, it is vital to take advantage of the role of bilateral and multilateral trade agreements that Vietnam has signed by expanding markets to partner countries (such as EVFTA and TPTPP) to create attractive conditions for investors from those countries.

Third, it will be important to review policy implementation to avoid overlap between recent documents and the earlier documents that remain valid. Moreover, it is critical to propose recommendations to the relevant authorities to develop and perfect investment policies synchronously and consistently and to improve administrative procedures continuously to create conditions favourable for foreign investors in the agricultural sector.

Choosing the entry mode is a topic addressed in many studies. However, due to the complexity of international business activities, many issues are still not explained in a unified manner, especially in different research contexts. Therefore, it is necessary to conduct more studies on investment attraction for each geographical area and with similar commercial relationships. One of the significant limitations of this study is that the primary data used were collected from secondary data. The data ensure objectivity but restrain the factors included in the research model. Therefore, it is necessary to encourage studies that consider additional factors such as international experience, investment motives, and technology level to provide accurate prediction results for choosing the suitable investment form.

**Supplementary Materials:** The following supporting information can be downloaded at: https://www.mdpi.com/article/10.3390/su14063164/s1, Table S1: Descriptive Statistics Information; Table S2: Descriptive Statistics Result; Table S3: One-way ANOVA by Region Information; Table S4: One-way ANOVA by Region Result; Table S5: One-way ANOVA by Region Result (cont); Table S6: Levene Statistic by Region Result; Table S7: Multiple Comparisons by Region; Figure S1:

Mean of InDis by Region; Figure S2: Mean of CulDis by Region; Figure S3: Mean of LnGeo by Region; Figure S4: Mean of LnEcoDis by Region; Table S8: One-way ANOVA by Investment Period Information; Table S9: One-way ANOVA by Investment Period Result; Table S10: One-way ANOVA by Investment Period Result (cont); Table S11: Levene Statistic for Investment PeriodResult; Table S12: Multiple Comparisons by Investment Period; Figure S5: Mean of InDis by Investment Period; Figure S6: Mean of CulDis by Investment Period; Figure S7: Mean of LnGeoDis by Investment Period; Figure S8: Mean of LnEcoDis by Investment Period; Table S13: Correlations Information; Table S14: Correlations Result; Table S15: Correlations Result (Cont); Table S16: Model 2 Information; Table S17: Variables Entered Method; Table S18: Model Summary; Table S19: Variables ANOVA; Table S20: Coefficients; Table S21: VIF Result; Table S22: Collinearity Diagnostics; Table S23: Collinearity Diagnostics (Cont); Table S24: Model 2 Logistic Regression; Table S25: Model 2 Case Processing Summary; Table S26: Model 2 Case Processing Summary; Table S27: Model 2 Iteration History—Beginning Block; Table S28: Model 2 Classification Table—Beginning Block; Table S29: Model 2 Variables in the Equation—Beginning Block; Table S30: Model 2 Variables not in the Equation—Beginning Block; Table S31: Model 2 Iteration History–Enter; Table S32: Model 2 Iteration History–Enter; Table S33: Model 2 Model Summary—Enter; Table S34: Model 2 Hosmer and Lemeshow Test—Enter; Table S35: Model 2 Contingency Table for Hosmer and Lemeshow Test—Enter; Table S36: Model 2 Classification Table—Enter; Table S37: Model 2 Variables in the Equation—Enter; Table S38: Model 1—Logistic Regression; Table S39: Model 1 Case Processing Summary; Table S40: Model 1Dependent Variable Encoding; Table S41: Model 1Iteration History—Beginning Block; Table S42: Model 1 Classification Table—Beginning Block; Table S43: Model 1 Variables in the Equation—Beginning Block; Table S44: Model 1 Variables not in the Equation- Beginning Block; Table S45: Model 1 Iteration History- Enter; Table S46: Model 1 Omnibus Tests of Model Coefficients- Enter; Table S47: Model 1 Model Summary- Enter; Table S48: Model 1 Hosmer and Lemeshow Test—Enter; Table S49: Model 1 Contingency Table for Hosmer and Lemeshow Test—Enter; Table S50: Model 1 Classification Table- Enter; Table S51: Model 1 Variables in the Equation- Enter; Table S52: Data of Cultural Index; Table S53: Data of FDI enterprises in the Vietnam's agricultural sector.

**Author Contributions:** Conceptualization, N.N. and T.D.V.; methodology, N.N. and T.D.V.; validation, N.N. and T.D.V.; formal analysis, N.N. and T.D.V.; resources, N.N.; data curation, N.N.; writing—original draft preparation, H.T.H.T.; writing—review and editing, N.N., H.T.H.T., and T.D.V. All authors have read and agreed to the published version of the manuscript.

**Funding:** This research received no external funding.

**Institutional Review Board Statement:** Not applicable.

**Informed Consent Statement:** Informed consent was obtained from all subjects involved in the study.

**Data Availability Statement:** Not applicable.

**Conflicts of Interest:** The authors declare no conflict of interest.

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
