# Peer review of "Country Distance and Entry Mode Choice of MNEs in Vietnam’s Agricultural Sector in Context of Free Trade"

_sustainability, doi:10.3390/su14063164_

Round 1

Reviewer 1 Report

I have no major comments . The authors have chosen correct  research methods. The results are OK. The authors appropriate commented  the performed analyzes.  The conclusions  correspond to the aim of the article and the established hypotheses. 

Author Response

Thank you for giving us the opportunity to submit a revised draft of the manuscript “Country Distance and Entry Mode Choice of MNEs in Vietnam’s Agricultural Sector in Context of Free Trade” for publication in Sustainability.We appreciate the time and effort that you and the reviewers dedicated to providing feedback on our manuscript and are grateful for the insightful comments on and valuable improvements to our paper. We have incorporated most of the suggestions made by the reviewers. Those changes are highlighted within the manuscript. Please see the attachment!

Reviewer 2 Report

This is an interesting topic. I particularly liked 2.2.-2.7. sub-chapters as each provided a hypothesis at the end basd on the related literature. Practical implications could be further strengthened. Besides, I have a couple of small comments:

1, line 297 is unfinished

2, line 411 I would not use independence as that means 0.00 values.

3, line 416. Not in this study, just in this database or data sample.

4, line 436-437. This hypothesis was formulated in the opposite way earlier.

Author Response

(The authors gave the same response as above.)

Reviewer 3 Report

The paper focuses on a really interesting field of the FDI flow: how the different distances are influencing the mode of entry. Which makes it more unique is that it is targeting the analysis of investments in Vietnamese agriculture.

The abstract is well written, comprehensive and compact, keywords are appropriate.

The introduction is well based, setting the research questions, the theoretical background, and the context.

The literature review is well written, detailed enough, provides a good picture of the different parts of the topic. Hypotheses are defined in the framework of the literature processing. However, in my evaluation, at the beginning of the literature review, a general overview of the literature of FDI flow in emerging countries would be needed. For that, I recommend some sources as follows: http://dx.doi.org/10.21511/ppm.18(1).2020.14 https://doi.org/10.21003/ea.V172-03 http://dx.doi.org/10.14254/2071-8330.2019/12-3/4 http://dx.doi.org/10.14254/2071-8330.2020/13-1/11

The methods are well defined and selected, which supports the research results and discussion well.

As a whole, the paper is a good piece and will contribute to the existing knowledge a lot.

Author Response

(The authors gave the same response as above.)
